# Overcoming Catastrophic Forgetting in Federated Class-Incremental Learning via Federated Global Twin Generator

## Abstract

Federated Class-Incremental Learning (FCIL) increasingly becomes essential in the decentralized setting, where it enables multiple participants to collaboratively train a global model to perform well on a sequence of tasks without sharing their private data. In FCIL, conventional Federated Learning algorithms such as FedAvg often suffer from catastrophic forgetting, resulting in significant performance declines on earlier tasks. Recent works based on generative models produce synthetic images to help mitigate this issue across all classes. However, these approaches' testing accuracy in previous classes is still much lower than recent classes, i.e., having better plasticity than stability. To overcome these issues, this paper presents Federated Global Twin Generator (FedGTG), an FCIL framework that exploits generative-model training on the global side without accessing client data. Specifically, the server trains a data generator and a feature generator to create two types of information from all seen classes. Then, it sends the synthetic data to the client. The clients then use feature-direction-controlling losses to make the local models retain knowledge and learn new tasks well. We extensively analyze the robustness of FedGTG on natural images and its ability to converge to flat local minima and achieve better predicting confidence (calibration). Experimental results on CIFAR-10, CIFAR-100, and tiny-ImageNet demonstrate the improvements in accuracy and forgetting measures of FedGTG as well as the robustness of domain shifts compared to previous frameworks.

## 1 Introduction

Federated Learning (FL) (McMahan et al., 2016; Bonawitz et al., 2019) is a Machine Learning setting that facilitates collaborative learning while maintaining privacy. Despite its significant achievements on various domains (Doshi & Yilmaz, 2022; Lin et al., 2021; Liu & Yang, 2021; Nguyen et al., 2019), FL observes several critical challenges, including resource limitation and data heterogeneity. Moreover, the client's local data distribution is assumed to remain unchanged, but the real-world scenarios (Aljundi, 2019) can be different, where the client's task, data, and domain can be changed. To overcome such challenges, Federated Class-Incremental Learning (FCIL) (Dong et al., 2022; 2023) is an innovative approach that combines the principles of FL and Class-Incremental Learning (CIL) (Rebuffi et al., 2017) to enable models to learn continuously from decentralized data sources while adapting to new information over time without forgetting previous knowledge (French, 1999). This approach addresses data privacy challenges and ensures the model can evolve as new data types or tasks emerge without accessing historical data. In CIL, exemplar-based approaches (Rebuffi et al., 2017; Chaudhry et al., 2019; Buzzega et al., 2020) preserve a limited number of samples from previous tasks to prevent forgetting, whereas exemplar-free approaches (He et al., 2018; Liu et al., 2020; Magistri et al., 2024) do not retain any samples from prior tasks.

In the FL setting, where privacy issues pose significant challenges, the exemplar-free category is particularly interesting since users cannot store historical data. Recent works in this field, such as TARGET (Zhang et al., 2023), FedCIL (Qi et al., 2023) and MFCL (Babakniya et al., 2024) tend to generate synthetic data and combine with distilling regularizers (Hinton et al., 2015; Liu et al., 2020) to balance the trade-off between retaining knowledge and learning new tasks. However, experimental results have shown that these works still witness catastrophic forgetting, i.e., bias

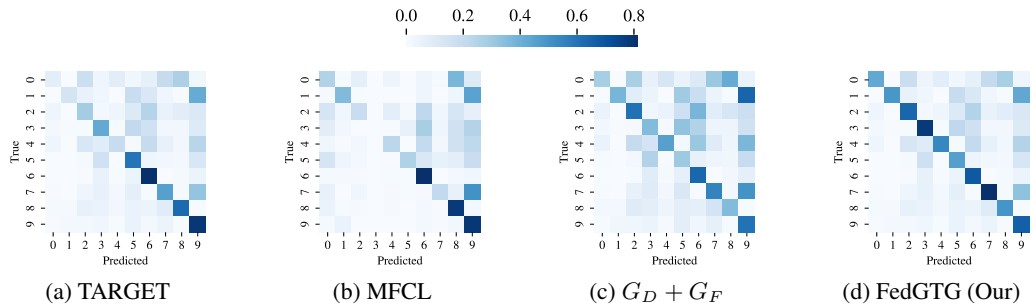

(a) TARGET      (b) MFCL      (c) $G_D + G_F$      (d) FedGTG (Our)

Figure 1: Confusion matrix among FCIL algorithms: **(a)** TARGET, **(b)** MFCL, **(c)** only the application of two generators to FL, and **(d)** FedGTG, testing on CIFAR-10 after training is completed. While TARGET and MFCL have bad predicting performance on initial classes and two generators struggle to learn new tasks, FedGTG achieves a better stability-plasticity trade-off.

towards recent classes, as shown Figure 1a and 1b. This is because the model trained by MFCL and TARGET is closely linked to its data generator. When the model begins to lose previous knowledge, it impacts the data generator, leading to the production of poor-quality synthetic images of earlier classes. Consequently, the model's performance on these classes in later tasks will significantly drop (Babakniya et al., 2024).

To address this problem, we propose **Federated Global Twin Generator** (FedGTG), an FCIL framework that does not store client data. Specifically, after completing one task, the server trains two generative models and shares them with clients on subsequent tasks to create synthetic examples and features of previous classes. On the client side, we propose a synthetic logit distillation using generated features for retaining knowledge and a fine-tuning loss using both real and generated data to be able to predict all classes. However, using only these two objectives still hinders the model's ability to obtain new knowledge, as shown in Figure 1c. We argue that this issue happened as the feature directions were not constrained. Therefore, we add a feature-direction-controlling loss, which helps the model have more plasticity in learning new tasks. As a result, FedGTG outperforms previous methos in accuracy and forgetting measures as shown in Figure 1d and Section 4.2.

We summarize our contributions below:

- We propose an FCIL framework that trains a data generator and a feature generator on the server side. These generators are distributed to the clients to mitigate forgetting.

- To help the model have a better stability-plasticity trade-off, we propose direction-controlling objectives on the client side.

- We conducted extensive experiments to demonstrate the effectiveness of our method in popular benchmarks and handling domain shifts.

- Moreover, we analyze the robustness of FedGTG compared with recent FCIL algorithms on natural images, its abilities to converge to flat minima, achieve better predicting confidence, and maintaining the effectiveness across different client sizes.

## 2 RELATED WORK

### 2.1 CONTINUAL LEARNING

Catastrophic forgetting (French, 1999) is a significant issue in machine learning where training a model on new data makes it forget previously learned knowledge. This issue is central to the field of CL, whose primary objective is to build models to acquire new knowledge while retaining information from older tasks. Numerous strategies have been explored to mitigate this problem, including the implementation of regularization terms (Li & Hoiem, 2017; Kirkpatrick et al., 2017; Zenke et al., 2017), the isolation of architectural parameters (Mallya & Lazebnik, 2018; Yoon et al., 2017; Mallya et al., 2018), the use of storing prior data (Rebuffi et al., 2017; Chaudhry et al., 2019; Buzzega et al., 2020), and studies of data generation (He et al., 2018; Zhuang et al., 2022; 2023; 2024; Magistri

et al., 2024). In CL, replay-based methods observe significant performance in accuracy and forgetting measures. However, privacy concerns in FL prevent data storage (Dong et al., 2022), making these methods inapplicable. An extensive alternative to address this issue is generative-based approaches.

These mitigation strategies become more crucial depending on the type of learning in CL. Specifically, there are three main types: Task-Incremental Learning (Task-IL), Domain-Incremental Learning (Domain-IL), and Class-Incremental Learning (Class-IL) (Van de Ven & Tolias, 2019). Each task is distinct in Task-IL and comes with its distribution during training and testing. In Domain-IL, the learning task does not change, while different domains or data distributions sequentially arrive during training. In Class-IL, each new task adds classes to what the model has to learn, continually expanding the amount of information the model needs to handle.

## 2.2 Federated Class-Incremental Learning

FCIL aims to train a model to learn new classes over time without forgetting previously learned classes while ensuring that data privacy is maintained across multiple decentralized devices. Several approaches exploit Knowledge Distillation (Hinton et al., 2015) to mitigate forgetting by appointing the global model's weight of the most recent task as a teacher. **Continual Federated Learning with Distillation (CFeD)** (Ma et al., 2022) constructs server and client-side knowledge distillation using a surrogate dataset, but this costs time and financial resources to collect enough data for this surrogate dataset. **Global-Local Forgetting Compensation (GLFC)** (Dong et al., 2022) relaxes this problem by training a proxy server globally to ease the imbalance issue between classes.

The above approaches yield extensive performance in past knowledge retention but cannot learn well on new tasks. To alleviate this issue, **Federated Class-Incremental Learning (FedCIL)** (Qi et al., 2023) trains generators at both client-side and server-side, as well as utilizing knowledge distillation to balance the stability-plasticity trade-off. However, this approach raised a privacy risk since information about client-side generative models is shared with the server. **Federated Class-Continual Learning via Exemplar-Free Distillation (TARGET)** (Zhang et al., 2023) and **Mimicking Federated Continual Learning (MFCL)** (Babakniya et al., 2024) relax this issue by training a data generator on the global-side and adding distilling regularizers to the client-side training to enhance overall performance. However, as shown in Figure 1, these methods still perform badly on old classes, leaving catastrophic forgetting mitigation a desirable goal.

## 3 Methodology

### 3.1 Preliminaries

There are $c$ clients and a central server, denoted as $\{C_1, C_2, \ldots, C_c\}$ and $S$, respectively. We consider the Synchronous Federated Continual Learning setting (Yang et al., 2024) where all clients share the same task sequence $T = \{T_1, T_2, \ldots, T_n\}$. At task $T_t$, each client $C_i$ has a private dataset $\mathcal{D}_i^t = \{\mathcal{X}_i^t, \mathcal{Y}^t\}$. During the first task, the global model $\theta_G^1$ is obtained after aggregating local models $\{\theta_1^1, \theta_2^1, \ldots, \theta_{s_1}^1\}$ using conventional Deep Learning methods, where $s_1$ is a number of selected clients among all. From the task $T_t$, $t \geq 2$, the global model $\theta_G^{t-1}$ can distinguish the samples belonged to the classes set $\bigcup_{i=1}^{t-1} \mathcal{Y}_i$. The server then distributes its parameters back to the clients. Client $C_i$ uses $\theta_G^{t-1}$ as an initial model to train on task $T_t$ using its private dataset $\mathcal{D}_i^t$. The local model $\theta_i^t$ should perform well in classifying classes from the set $\bigcup_{i=1}^{t} \mathcal{Y}_i$. Finally, the server collects the local models from clients who participate in the process after each $r_t$ communication round and obtains a new global model $\theta_G^t$, which can identify classes from the set $\bigcup_{i=1}^{t} \mathcal{Y}_i$.

### 3.2 Overview

Several replay-based approaches (Rebuffi et al., 2017; Chaudhry et al., 2019; Buzzega et al., 2020) in the conventional CL achieve significant performance across all classes by storing a subset of samples from previous tasks. However, these methods are not viable in the FL setting due to privacy concerns (e.g., local hospitals cannot share data with the central server). One initial solution is using generative models, which can generate synthetic data for subsequent training, as demonstrated in earlier studies (Zhang et al., 2023; Babakniya et al., 2024). However, only generating synthetic examples causes

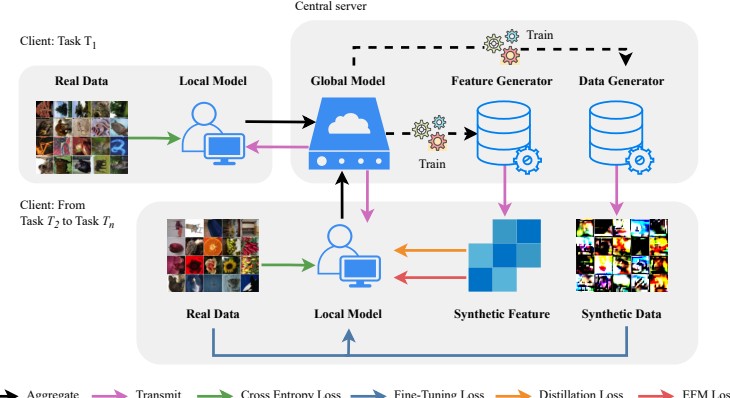

Figure 2: Illustration of the proposed framework. After completing one task, the server employs a data-free approach to train two generators. The clients then use two types of synthetic information from these generators to train their local models for retaining knowledge and learning new tasks well.

forgetting in previously learned classes, as shown in Figure 1. This is because synthetic images cannot fully reflect the knowledge from prior works (Liu et al., 2020). The authors also show that synthetic features can alleviate this problem by capturing the information held within the model's weights. Therefore, we also train a feature generator in addition to the data generator. Specifically, we propose a Federated Global Twin Generator (FedGTG), which can balance the stability-plasticity trade-off. This method has two main stages: (1) At the end of each task, the server trains a data generator and a feature generator to capture the information of all seen classes; (2) Clients receive generators from the server to create synthetic information, and obtaining global weights as initialization, which helps retain knowledge from previous tasks and learn the new task efficiently.

### 3.3 SERVER-SIDE DATA GENERATOR

Since the server only has access to the global model's weights, we can only train the data generator using data-free methods, such as DeepInversion (Yin et al., 2020). Specifically, we have a generative model that takes a noise $z \sim \mathcal{N}(0, 1)$ as input and produces a synthetic example $\tilde{x}$ mirroring the dimensions of the original training input. This synthetic data should observe the following objectives.

After training task $T_t$, the synthetic data should be classified correctly by the global model $\theta_G^t$ and not be biased to any classes. With this aim, we employ a modified cross entropy classification loss between its assigned label $z$ and the prediction of $\theta_G^t$ on $G_D^t(z)$, as follows:

$$\mathcal{L}_{\text{CE}} = \text{CE}_{\text{last}}\left(\text{argmax}\left(z\left[:,q\right]\right), \theta_G^t\left(\widetilde{x}\right) + \lambda_{\text{current}}\text{CE}_{\text{current}}\left(\text{argmax}\left(z\left[:,q\right]\right), \theta_G^t\left(\widetilde{x}\right)\right), \quad (1)$$

where $\widetilde{x}$ is the output of $G_D^t(z)$; $q$ is the total number of classes seen in the previous tasks, we just take $q$ dimension for the noise; $\text{CE}_{\text{last}}$ and $\text{CE}_{\text{current}}$ respectively are the Cross-Entropy Loss using the truncated outputs of $\theta_G^t(\widetilde{x})$ corresponding with last classes from task $T_i$, $i < t$, and current learned classes on task $T_t$, and $\lambda_{\text{current}}$ is the temperature hyper-parameter.

Generating synthetic examples can easily be biased to a subset of classes. To maintain the diversity between classes, we utilize the Information Entropy (IE) Loss (Chen et al., 2019) as follows:

$$\mathcal{L}_{\text{IE}} = -\mathbf{H}_{\text{info}}\left(\frac{1}{\text{bs}}\sum_{i=1}^{\text{bs}}\theta_G^t\left(\widetilde{x}_i\right)\right), \text{bs: batch size}, \quad (2)$$

This loss measures the IE for samples of a batch. Maximizing this value can promote a more uniform and balanced output distribution from the generator across all classes.

To further improve the stability of generator training, we use Batch Normalization Loss (Smith et al., 2021) to make all Batch Normalization layers have the same statistics on synthetic images, as follows:

$$\mathcal{L}_{\text{batch}} = \frac{1}{L}\sum_{j=1}^{L}\mathbf{KL}\left(\mathcal{N}\left(\mu_j, \sigma_j^2\right) \| \mathcal{N}\left(\widetilde{\mu}_j, \widetilde{\sigma}_j^2\right)\right), \quad (3)$$

where $L$ is the total number of Batch Normalization layers in the architecture of the global model. $\mu_j$ and $\sigma_j^2$ are the mean and variance stored in Batch Normalization layer $j$ of the global model, $\widetilde{\mu}_j$ and $\widetilde{\sigma}_j$ are measured statistics of Batch Normalization layer $j$ for the synthetic data.

Adjacent pixels in real images typically have values near one another. One typical method to promote similar patterns in the synthetic images is to add Image Prior Loss (Haroush et al., 2020). We can create the smoothed (blurred) version of an image by applying a Gaussian kernel and minimizing the distance of the original and Smooth $(\widetilde{x})$ as

$$\mathcal{L}_{\text{smooth}} = \|\widetilde{x} - \text{Smooth}\,(\widetilde{x})\|_2^2. \tag{4}$$

In summary, we can write the training objective of $G_D$ as follows:

$$\min_{G_D} \mathcal{L}_{\text{CE}} + \lambda_{\text{IE}}\mathcal{L}_{\text{IE}} + \lambda_{\text{batch}}\mathcal{L}_{\text{batch}} + \lambda_{\text{smooth}}\mathcal{L}_{\text{smooth}}, \tag{5}$$

where $\lambda_{\text{IE}}$, $\lambda_{\text{batch}}$, and $\lambda_{\text{smooth}}$ are hyper-parameters of specific loss functions.

## 3.4 SERVER-SIDE FEATURE GENERATOR

As mentioned in Section 3.2, only synthetic images can exacerbate the catastrophic forgetting problem. (Liu et al., 2020) has shown that features can store better knowledge of previous tasks than data. Therefore, we train a feature generator that synthesizes features, capturing the knowledge within the feature space. Like the data generator, this generative model is trained only on the server side. The feature generator takes noise input $z \sim \mathcal{N}(0, 1)$ and produces synthetic features $\tilde{f}$ that match the dimensions of the original features. These synthetic features must meet the following objectives:

After training task $T_t$, the generative feature should be classified correctly by the classifier $H_G^t$ of the global model. Additionally, the synthetic features should not be biased to any classes. With this aim, we employ a temperature cross entropy classification loss between its assigned label $z$ and the prediction of $H_G^t$ on $G_D^t(z)$ as

$$\mathcal{L}_{\text{FCE}} = \text{CE}_{\text{last}}\left(\text{argmax}\,(z\,[:,q])\,, H_G^t\left(\widetilde{f}\right)\right) + \lambda_{\text{current}}\text{CE}_{\text{current}}\left(\text{argmax}\,(z\,[:,q])\,, H_G^t\left(\widetilde{f}\right)\right), \tag{6}$$

where $\widetilde{f}$ is the output of $G_F^t(z)$; $q$ is the total number of classes seen in the previous tasks, we take $q$ dimension for the noise; $\text{CE}_{\text{last}}$ and $\text{CE}_{\text{current}}$ respectively are the Cross-Entropy Loss using the truncated outputs of $H_G^t\left(\widetilde{f}\right)$ corresponding with last classes from task $T_i$, $i < t$, and current learned classes on task $T_t$, and $\lambda_{\text{current}}$ is the temperature hyper-parameter.

The generated features should not be biased to any subset of classes. Therefore, we propose the Feature Information Entropy Loss to make the synthetic feature have this quality, which is

$$\mathcal{L}_{\text{FIE}} = -\mathbf{H}_{\text{info}}\left(\frac{1}{\text{bs}}\sum_{i=1}^{\text{bs}} H_G^t\left(\widetilde{f_i}\right)\right), \text{bs: batch size}, \tag{7}$$

In summary, we can write the training objective of $G_F$ as follows:

$$\min_{G_F} \mathcal{L}_{\text{FCE}} + \lambda_{\text{FIE}}\mathcal{L}_{\text{FIE}}, \tag{8}$$

where $\lambda_{\text{FIE}}$ is the hyper-parameter of Feature Information Entropy Loss.

## 3.5 CLIENT-SIDE

For the first task, clients will carry out the traditional FL process after receiving the global model weights from the server. For each subsequent task, clients will use the data and feature generators provided by the server to produce synthetic information throughout the task. Note that the transmission of these generators from the server occurs only once per task. Specifically, from the second task onward, the local models need to learn the current task quickly, as well as retaining knowledge from previous tasks efficiently. Therefore, we divide the learning objectives into two parts, as follows:

To learn new tasks well, the model needs to learn the new information separately from the old classes. We compute the Cross-Entropy Loss using only the new classes' linear heads. Formally, we minimize:

$$\mathcal{L}_{\text{CE}} = \text{CE}\left(\theta^t\left(x \mid T_t\right), y\right), \tag{9}$$

where $\theta^t\left(x \mid T_t\right)$ is model's output and masking old classes before task $T_t$'s linear heads.

To mitigate forgetting, previous approaches leverage knowledge distillation (Usmanova et al., 2021; Zhang et al., 2023). However, this can cause information loss in probability space due to squashing functions (Liu et al., 2018). Therefore, motivated by (Buzzega et al., 2020), we propose Synthetic Logits Distillation Loss, which matches the logits of the old and current linear heads. These classifiers take synthetic features as input instead of synthetic data since the feature stores more previous information. Formally, we optimize:

$$\mathcal{L}_{\text{logits}} = \left\| \theta^{t-1}_G\left(\widetilde{f}\right) - \theta^t\left(\widetilde{f}\right) \right\|, \tag{10}$$

where $\theta^{t-1}_G$ is the global model trained up to task $T_{t-1}$.

As shown in (Babakniya et al., 2024), when there is a sudden shift in the distribution of the input of the task sequence, biased features on previous tasks can output biased logits, hindering the ability to obtain new knowledge. To mitigate this shortcoming, we utilize only the extracted features of the data, i.e., clients freeze the feature extraction layers and update only the linear head (represented by $H^t$) for both real ($x$) and synthetic ($\widetilde{x}$) images. This Fine-tuning loss is formulated as

$$\mathcal{L}_{\text{FT}} = \text{CE}\left(H^t\left(\left[f, \widetilde{f}\right]\right), [y, \widetilde{y}]\right), \tag{11}$$

where $f$ and $\widetilde{f}$ respectively are the extracted features of $x$ and $\widetilde{x}$ after passing through the freezed feature extractor $F^t$ of the local model, $y$ and $\widetilde{y}$ is the hard label of $x$ and $\widetilde{x}$.

Figure 1c shows that combining the above objectives reduces the model's performance across all classes. We contend that this happens because the feature directions are unconstrained, resulting in the total loss failing to converge. We then add additional loss to balance this problem, named Empirical Feature Matrix Loss (Magistri et al., 2024), which constrains directions in feature space most important for previous tasks. At the same time, it allows more plasticity in other directions when learning new tasks. In this work, we re-utilize the synthetic features to calculate the Empirical Feature Matrix $E_{t-1}$ from the previous task $T_{t-1}$. We have,

$$\mathcal{L}_{\text{EFM}} = \left(F^t\left(x\right) - F^{t-1}_G\left(x\right)\right)^\top \left(\lambda_E E_{t-1} + \eta I\right)\left(F^t\left(x\right) - F^{t-1}_G\left(x\right)\right), \tag{12}$$

where $F^t$ and $F^{t-1}_G$ respectively are the feature extractor of the current model and the previous global model, $\eta$ is the damping term to constrain features to stay in a specific region.

In summary, the final objective on the client side as

$$\min_{\theta^t} \mathcal{L}_{\text{CE}} + \lambda_{\text{logits}}\mathcal{L}_{\text{logits}} + \lambda_{\text{FT}}\mathcal{L}_{\text{FT}} + \lambda_{\text{EFM}}\mathcal{L}_{\text{EFM}}. \tag{13}$$

## 4 EXPERIMENTAL RESULTS

### 4.1 EXPERIMENTAL SETUP

**Datasets** We perform our experiments on three widely-used benchmark datasets in FCIL (Dong et al., 2022; Zhang et al., 2023; Babakniya et al., 2024), which are the protocol versions of **CIFAR-10** (Krizhevsky et al., 2009), **CIFAR-100** (Krizhevsky et al., 2009), **tiny-ImageNet** (Yao & Miller, 2015), and we name it respectively are **Sequential F-CIFAR-10**, **Sequential F-CIFAR-100** and **Sequential F-tiny-ImageNet**. Moreover, FCIL is usually applied in the finance and healthcare industries (Wang et al., 2024), where the data distribution shifts significantly. We want to investigate the effectiveness of our work on this application. We introduce the protocol dataset named **HealthMNIST** to assess the domain shift scenario, which involved two distinct classification tasks: **Task 1** is the Colon Pathology Classification from PathMNIST (Yang et al., 2023) and **Task 2** is the Blood Cell Classification from BloodMNIST (Yang et al., 2023). The data preparation is explained later in Appendix B. We use Latent Dirichlet Allocation (LDA) (Reddi et al., 2020) with $\alpha = 1$ and $\alpha = 0.5$ to distribute the data of each task among clients. Additional experiments on **SuperImageNet** (Babakniya et al., 2024) are then provided in the Appendix D.3.

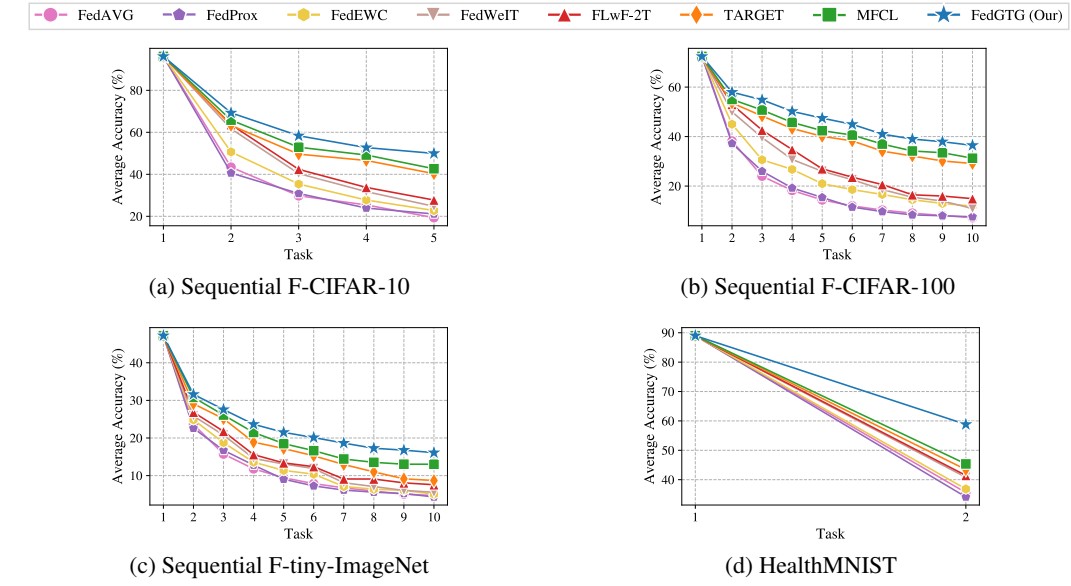

Figure 3: AA per task of various algorithms on several benchmarks under the IID scenario.

**FCIL Baselines**  We compare FedGTG with two conventional aggregating methods **FedAvg** (McMahan et al., 2017) and **FedProx** (Li et al., 2020), as well as three regularization-based FCIL methods, **FLwF-2T** (Usmanova et al., 2021) and the FCIL version of **FedWeIT** (Yoon et al., 2021) and **FedEWC** (Zhang et al., 2023), and two generative-based methods, **TARGET** (Zhang et al., 2023) and **MFCL** (Babakniya et al., 2024). The detailed description can be seen in Appendix B.

**Models and Implementation Details**  In all experiments, we train a ResNet-18 (He et al., 2016) backbone using the SGD optimizer (Bottou, 1998). We train the model for 100 epochs per task on every dataset. We use FedAvg (McMahan et al., 2017) for aggregation. Additional implementation detailsare then provided in the Appendix B. We also conducted experiments on different architectures, including ResNet-34 and ResNet-50, which can be found in the Appendix D.2.

**Evaluation Metrics**  We report the performance of the methods using two metrics: Average Incremental Accuracy and Average Forgetting. **Average Incremental Accuracy (AIA)** measures the average accuracy of the global model on all tasks after the training finishes. Forgetting $(f_t)$ of task $T_t$ is the difference between the model's best performance on task $T_t$ and its accuracy after completed training. Consequently, **Average Forgetting (AF)** is the average of all $f^t$, from task $T_1$ to task $T_{n-1}$, at the end of task $T_n$. We report the averaged result over three different random initializations.

## 4.2 PERFORMANCE RESULTS

We present the performance of FedGTG and the baselines. Figure 3 shows the Average Accuracy of the model at each task in the training process. It can be seen that FedGTG achieves state-of-the-art performance in all settings. Specifically, our method observes better Average Accuracy on all later tasks. Table 1 reports both AIA results *(higher is better)* and AF results *(lower is better)* under the IID and Non-IID data distribution, respectively.

As expected, FedAvg and FedProx suffer the highest forgetting since they are not designed for FCIL. Compared to FedEWC, FedWeIT and FLwF-2T, the performance gap between it and our FedGTG is significant, indicating that regularization towards previous parameter sets is insufficient to avoid forgetting. Compared to the generative-replay methods TARGET and MFCL, our FedGTG achieves the least AF and the best AIA, showing that FedGTG can effectively retain knowledge and learn new tasks. Moreover, FedGTG performs extensively in the context of domain shift, which can retain knowledge from the first task of HealthMNIST, where MFCL and TARGET fail to do so.

Table 1: Performance of the different baselines in terms of AIA and AF for four datasets. $\alpha = 1$ is the IID scenario, and $\alpha = 0.5$ is the Non-IID scenario. [↑] higher is better, [↓] lower is better.

| Method | CIFAR-10 | | CIFAR-100 | | tiny-ImageNet | | HealthMNIST | |
|---|---|---|---|---|---|---|---|---|
| | AIA (↑) | AF (↓) | AIA (↑) | AF (↓) | AIA (↑) | AF (↓) | AIA (↑) | AF (↓) |
| $\alpha = 1$ | | | | | | | | |
| FedAVG | $42.82 \pm 0.23$ | $55.55 \pm 0.58$ | $21.39 \pm 0.22$ | $78.67 \pm 0.83$ | $13.80 \pm 0.19$ | $74.12 \pm 0.81$ | $62.31 \pm 0.14$ | $43.27 \pm 0.75$ |
| FedProx | $42.43 \pm 0.32$ | $56.15 \pm 0.71$ | $21.54 \pm 0.32$ | $78.12 \pm 0.71$ | $13.69 \pm 0.21$ | $75.16 \pm 0.79$ | $61.56 \pm 0.24$ | $43.27 \pm 0.70$ |
| FedEWC | $45.27 \pm 0.17$ | $49.46 \pm 0.76$ | $26.63 \pm 0.29$ | $62.17 \pm 0.49$ | $14.58 \pm 0.15$ | $58.00 \pm 0.51$ | $62.92 \pm 0.16$ | $40.42 \pm 0.62$ |
| FedWeIT | $50.46 \pm 0.21$ | $45.99 \pm 0.59$ | $30.19 \pm 0.19$ | $55.57 \pm 0.48$ | $16.02 \pm 0.22$ | $46.23 \pm 0.77$ | $64.91 \pm 0.12$ | $39.59 \pm 0.52$ |
| FLwF-2T | $52.74 \pm 0.23$ | $39.51 \pm 0.59$ | $32.19 \pm 0.18$ | $50.78 \pm 0.63$ | $17.18 \pm 0.17$ | $44.51 \pm 0.67$ | $65.22 \pm 0.12$ | $36.92 \pm 0.69$ |
| TARGET | $59.19 \pm 0.16$ | $17.23 \pm 0.45$ | $42.15 \pm 0.13$ | $26.45 \pm 0.61$ | $19.46 \pm 0.25$ | $20.17 \pm 0.57$ | $66.23 \pm 0.22$ | $36.41 \pm 0.57$ |
| MFCL | $61.34 \pm 0.21$ | $22.32 \pm 0.52$ | $45.07 \pm 0.12$ | $28.30 \pm 0.78$ | $21.47 \pm 0.15$ | $23.90 \pm 0.58$ | $67.18 \pm 0.15$ | $36.23 \pm 0.52$ |
| **FedGTG (ours)** | $\mathbf{64.50 \pm 0.22}$ | $\mathbf{13.14 \pm 0.67}$ | $\mathbf{46.42 \pm 0.18}$ | $\mathbf{18.66 \pm 0.76}$ | $\mathbf{24.04 \pm 0.23}$ | $\mathbf{16.18 \pm 0.62}$ | $\mathbf{73.91 \pm 0.19}$ | $\mathbf{19.15 \pm 0.60}$ |
| $\alpha = 0.5$ | | | | | | | | |
| FedAVG | $40.92 \pm 0.26$ | $55.59 \pm 0.58$ | $20.66 \pm 0.25$ | $61.34 \pm 0.72$ | $11.82 \pm 0.22$ | $74.16 \pm 0.68$ | $59.93 \pm 0.14$ | $43.21 \pm 0.85$ |
| FedProx | $40.43 \pm 0.32$ | $55.15 \pm 0.67$ | $20.43 \pm 0.22$ | $62.73 \pm 0.81$ | $11.49 \pm 0.21$ | $75.01 \pm 0.72$ | $60.15 \pm 0.24$ | $43.21 \pm 0.12$ |
| FedEWC | $43.22 \pm 0.17$ | $50.70 \pm 0.66$ | $25.53 \pm 0.18$ | $59.17 \pm 0.56$ | $12.90 \pm 0.15$ | $60.93 \pm 0.55$ | $60.88 \pm 0.22$ | $42.45 \pm 0.32$ |
| FedWeIT | $48.11 \pm 0.21$ | $47.34 \pm 0.46$ | $28.89 \pm 0.20$ | $56.11 \pm 0.63$ | $14.55 \pm 0.11$ | $49.76 \pm 0.49$ | $61.05 \pm 0.17$ | $42.09 \pm 0.55$ |
| FLwF-2T | $50.23 \pm 0.23$ | $40.21 \pm 0.51$ | $30.25 \pm 0.14$ | $53.72 \pm 0.46$ | $16.14 \pm 0.18$ | $44.59 \pm 0.67$ | $63.11 \pm 0.12$ | $39.72 \pm 0.59$ |
| TARGET | $56.19 \pm 0.19$ | $19.45 \pm 0.45$ | $41.03 \pm 0.13$ | $28.23 \pm 0.68$ | $18.46 \pm 0.25$ | $22.23 \pm 0.57$ | $64.03 \pm 0.23$ | $37.41 \pm 0.60$ |
| MFCL | $56.65 \pm 0.25$ | $18.34 \pm 0.59$ | $42.07 \pm 0.25$ | $30.30 \pm 0.59$ | $21.42 \pm 0.19$ | $21.02 \pm 0.58$ | $65.11 \pm 0.19$ | $36.13 \pm 0.52$ |
| **FedGTG (ours)** | $\mathbf{61.11 \pm 0.18}$ | $\mathbf{13.12 \pm 0.37}$ | $\mathbf{44.58 \pm 0.21}$ | $\mathbf{20.89 \pm 0.76}$ | $\mathbf{23.23 \pm 0.27}$ | $\mathbf{15.18 \pm 0.69}$ | $\mathbf{68.66 \pm 0.23}$ | $\mathbf{25.15 \pm 0.71}$ |

## 4.3 MODEL ANALYSIS

The majority of FCIL research concentrates on testing experiments on ideal benchmarks (Dong et al., 2022; Zhang et al., 2023; Babakniya et al., 2024), such as CIFAR (Krizhevsky et al., 2009) and ImageNet (Deng et al., 2009). This lacks analysis concerning real-world scenarios, such as the decision-making required in hospitals or the model's generalization to diverse environments. Therefore, in this section, we conducted experiments to analyze the robustness of FedGTG and three FCIL algorithms on corrupted environments, and the qualities of generalization (Chaudhari et al., 2019; Keskar et al., 2016) and achieve calibrated networks (Guo et al., 2017; Kull et al., 2019).

**Robustness to natural corruptions.** We evaluate our method and the recent FCIL methods on the **CIFAR-100-C** dataset. This dataset includes 18 augmentations of the original CIFAR-100, inspired by CIFAR-10-C (Hendrycks & Dietterich, 2019). Models are trained using standard CIFAR-100 with the same setting in Section 4.1 and tested on CIFAR-100-C. Figure 4 shows robustness to 09 different corruptions averaged over three different runs, the results of the rest augmentations are shown in Appendix D.4. Specifically, our approach achieves higher test accuracy on various corruptions, with an average improvement of 5% over MFCL and 8% over TARGET. Evidently, our method offers noticeable advantages in robustness against natural corruption.

**Converging to flatter minima.** Extensive CL algorithms (Bhat et al., 2022; Wang et al., 2023; Park et al., 2024) explore how well their methods generalize by examining their ability to converge to flat minima. In this part, we compare the flatness of the training minima of FLwF-2T, TARGET, and MFCL with FedGTG. As done in (Zhang et al., 2019), we consider the model at the end of training and add independent Gaussian noise with growing variance to each parameter. This allows us to evaluate its effect on the average loss $\sum_{t=1}^{n} \mathcal{L}_{CE}^{(T_t)}$ across all training examples. As shown in Figures 5a and 5b, MFCL, especially FLwF-2T, and TARGET, reveal higher sensitivity to perturbations than FedGTG. This concludes that FedGTG can achieve better generalization than previous methods.

**Converging to a more calibrated network.** Calibration measures how well a learner's prediction confidence matches its accuracy, with ideal outcomes reflecting true probabilities of correctness. In real-world applications, including weather forecasting (Bröcker, 2009) and econometric analysis (Gneiting et al., 2007), the calibrating ability of a model should be investigated. Figures 5c and 5d show the value of the Expected Calibration Error (ECE) (Naeini et al., 2015) across various FCIL methods after completing each task. It can be seen that FedGTG achieves a lower ECE than the others, proving that models trained with FedGTG are less over-confident and easier to interpret.

**Robustness to different client sizes.** We validate the effectiveness of FedGTG across different client sizes on the tiny-ImageNet dataset. We run experiments by varying the number of total clients

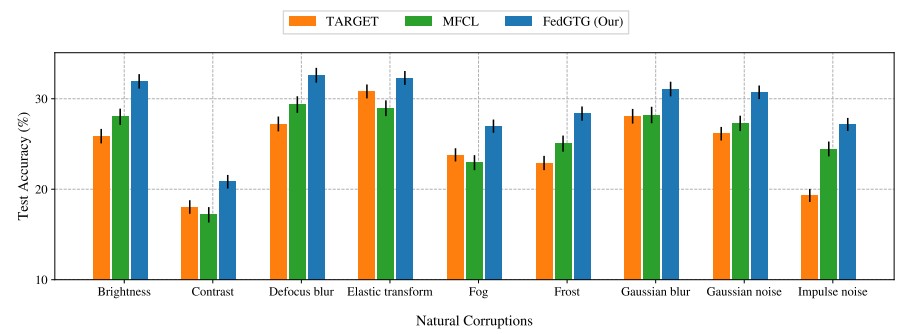

Figure 4: Robustness to natural corruptions.

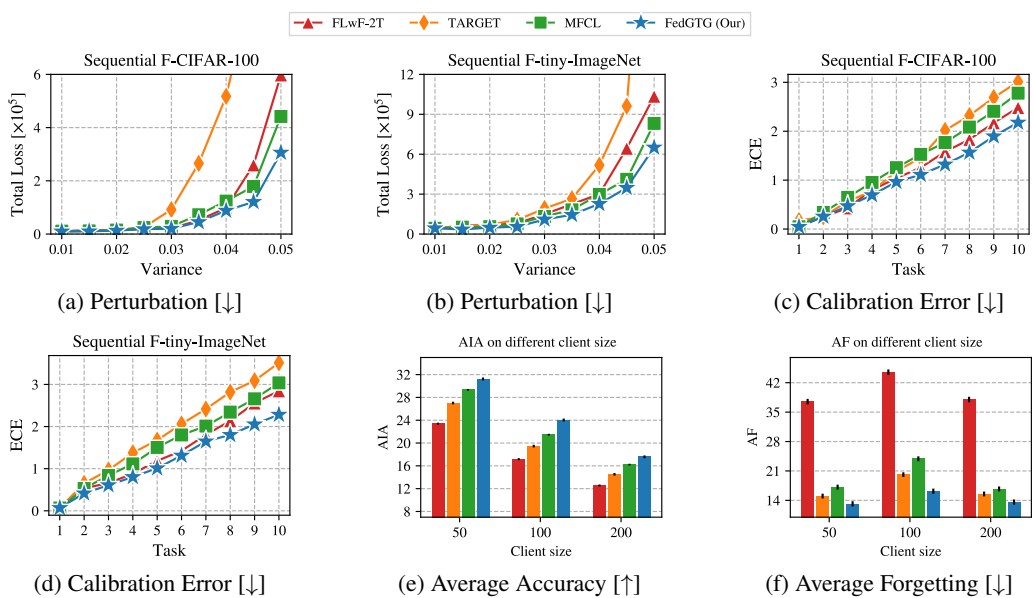

Figure 5: Results for the model analysis. [↑] higher is better, [↓] lower is better *(best seen in color)*.

(maintaining a consistent participation rate of $0.1$ per round), ranging from 50 to 200, and compare the results. Figures 5e and 5f demonstrate that our method still outperforms other approaches, achieving an accuracy 4% higher and a forgetting rate 6% lower compared to the next best method, MFCL.

## 5 ABLATION STUDY

We highlight the significance of each loss within our framework and analyze both server and client contributions by sequentially removing components to observe their effects. Table 2 shows our results, where each row corresponds to the removal of a specific loss component, and the columns display the corresponding Average Accuracy ($\mathcal{A}^t$), for $1 \leq t \leq 10$, Average Incremental Accuracy ($\mathcal{A}$), and Average Forgetting ($\mathcal{F}$). Specifically, the performance of the model is influenced by generative models, as poorly trained ones result in low AIA and high AF compared to others. Nevertheless, the Fine-tuning Loss has the lowest AF because it did not learn tasks well (lowest AIA). The final two rows illustrate how the feature-constraining losses ($\mathcal{L}_{\text{logits}}$ and $\mathcal{L}_{\text{EFM}}$) impact the performance of the global model, where the decrease in accuracy demonstrates the importance of these two losses.

## 6 DISCUSSION

Since two generative models are trained using the global model solely, the clients do not have to send their data to the server. Moreover, as shown in Appendix E, the visualization of synthetic images does

Table 2: Ablation study for FedGTG on Sequential F-CIFAR-100.

| w/o Loss | $\mathcal{A}^1$ | $\mathcal{A}^2$ | $\mathcal{A}^3$ | $\mathcal{A}^4$ | $\mathcal{A}^5$ | $\mathcal{A}^6$ | $\mathcal{A}^7$ | $\mathcal{A}^8$ | $\mathcal{A}^9$ | $\mathcal{A}^{10}$ | $\mathcal{A}$ | $\mathcal{F}$ |
|---|---|---|---|---|---|---|---|---|---|---|---|---|
| $\mathcal{L}_{\text{IE}}$ | 72.40 | 56.60 | 46.96 | 38.22 | 33.18 | 30.73 | 28.50 | 25.05 | 23.98 | 22.37 | 33.95 | 37.31 |
| $\mathcal{L}_{\text{batch}}$ | 72.40 | 55.62 | 48.67 | 41.25 | 33.48 | 30.08 | 27.10 | 22.98 | 22.48 | 21.41 | 33.67 | 41.42 |
| $\mathcal{L}_{\text{smooth}}$ | 72.40 | 57.15 | 49.10 | 41.85 | 39.44 | 37.58 | 34.67 | 31.15 | 30.50 | 29.20 | 38.96 | 29.36 |
| $\mathcal{L}_{\text{FIE}}$ | 72.40 | 56.35 | 47.96 | 37.22 | 34.18 | 32.61 | 29.82 | 26.17 | 24.98 | 23.03 | 34.70 | 33.42 |
| $\mathcal{L}_{\text{FT}}$ | 72.40 | 41.80 | 34.67 | 29.25 | 22.68 | 17.60 | 15.17 | 13.06 | 12.59 | 12.09 | 22.10 | 11.66 |
| $\mathcal{L}_{\text{logits}}$ | 72.40 | 57.15 | 49.77 | 43.05 | 39.36 | 37.72 | 35.67 | 33.19 | 32.73 | 31.62 | 40.03 | 25.35 |
| $\mathcal{L}_{\text{EFM}}$ | 72.40 | 56.30 | 48.77 | 42.53 | 39.56 | 37.28 | 35.37 | 31.91 | 31.66 | 29.92 | 39.26 | 27.38 |
| FedGTG (ours) | 72.40 | 57.95 | 54.85 | 50.20 | 47.45 | 44.95 | 41.00 | 39.01 | 37.92 | 36.46 | 46.42 | 18.66 |

not replicate any real data, and therefore, it will preserve privacy. In addition, our framework does not affect the aggregation stage, allowing the integration of Secure Aggregation techniques (Kim et al., 2023; Kanchan et al., 2024). This ensures that when local updates are sent to the server for aggregation, they remain encrypted, which prevents the server from accessing the client's information.

Table 3: Total parameters sent from the server to the clients across FCIL algorithms.

| Dataset/Method | FedGTG (ours) | MFCL | TARGET | FLwF-2T | FedWeIT | FedEWC | FedAvg | FedProx |
|---|---|---|---|---|---|---|---|---|
| CIFAR-10 | 20, 996, 877 | 19, 696, 397 | 19, 696, 397 | 11, 272, 458 | 11, 272, 458 | 11, 272, 458 | 11, 272, 458 | 11, 272, 458 |
| CIFAR-100 | 20, 949, 681 | 19, 649, 201 | 19, 649, 201 | 11, 225, 262 | 11, 225, 262 | 11, 225, 262 | 11, 225, 262 | 11, 225, 262 |
| tiny-ImageNet | 21, 416, 607 | 19, 853, 471 | 19, 853, 471 | 11, 281, 692 | 11, 281, 692 | 11, 281, 692 | 11, 281, 692 | 11, 281, 692 |
| HealthMNIST | 20, 949, 681 | 19, 649, 201 | 19, 649, 201 | 11, 225, 262 | 11, 225, 262 | 11, 225, 262 | 11, 225, 262 | 11, 225, 262 |

Table 4: Training time in seconds of different algorithms trained on the CIFAR-100 dataset.

| Training Time/Method | FedGTG (ours) | MFCL | TARGET | FLwF-2T | FedWeIT | FedEWC | FedProx | FedAvg |
|---|---|---|---|---|---|---|---|---|
| $t = 1$ | 1.2 | 1.2 | 1.2 | 1.2 | 1.2 | 1.2 | 1.8 | 1.2 |
| $t > 1$ | 4.1 | 3.7 | 3.5 | 3.4 | 2.2 | 1.2 | 1.8 | 1.2 |

In our work, the clients need to accommodate two generative models and the global model's weights from the most recent task, which introduces higher storage requirements than previous methods. However, this transmission of generators takes place **only once per task**, representing a necessary cost to prevent catastrophic forgetting. Table 3 reports the total parameters transmitted from the server to the clients, serving as a measure of the communication overheads. We also calculate the amount of time in seconds that the clients need to complete one round, as shown in Table 4. The implementational details of training time are provided in Appendix D.1. While FedGTG requires more parameters and time for training than others, it delivers significant benefits. As shown in Table 1 and Figure 3, FedGTG outperforms others in AIA and AF. Moreover, the framework proves effective in handling complex scenarios such as domain shifts, making the added computational cost justifiable, particularly in applications like healthcare and finance, where data privacy and performance are crucial. We can see that although FedGTG introduces additional computational components, these are essential to achieving a balance between retaining knowledge and learning new tasks in FCIL, which other methods struggle with.

## 7 CONCLUSION

In this work, we alleviate the lack of stability of previous works in the FCIL setting by introducing a framework named FedGTG, both utilizing data and feature generative models trained by the server, eliminating the requirement for costly on-device memory for clients. Our experiments show that FedGTG is successful in reducing catastrophic forgetting and surpasses the current state-of-the-art methods. Moreover, By analyzing the robustness on natural images, testing the qualities of converging to flat minima and calibrated networks, and the performance, as well as the performance on the context of domain shifts between FCIL algorihms, we observe that our framework outperforms previous approaches, making FedGTG more applicable in real-world scenarios.

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

## A    FEDGTG ALGORITHM

Recall that there are $n$ tasks $T_1, T_2, \ldots, T_n$. At task $T_1$, the system is trained using the conventional FedAvg algorithm for aggregating the weight from the clients in $R$ communication rounds. At the end of every task, the server trains a data generator and a feature generator without using any information from the clients. From task $T_2$, these two generators are sent to the clients, which combine with modified objectives to both retain knowledge and learn new tasks well. We formalize our approach in Algorithm 1 in detail.

---

**Algorithm 1** Federated Global Twin Generator

---

1: **Input:**
2: $n$ tasks with $n$ datasets $\{\mathcal{D}_1, \mathcal{D}_2, \ldots, \mathcal{D}_n\}$.
3: $c$ clients with $c$ local models $\theta$, $R$ communication rounds.
4: A global model $\theta_G$, a data generator $G_D$ and a feature generator $G_F$.
5: **Procedure:**
6: **for** $t = 1$ **to** $T$ **do**
7:     **for** $r = 1$ **to** $R$ **do**               $\triangleright$ Each task is learned on several communication rounds
8:         Select $k$ clients for training.
9:         **if** $r > 1$ or $t > 1$ **then**
10:             The server sends the global model's weight to selected clients.
11:             **if** $t > 1$ **then**
12:                 The server sends the two generators, the global model's weight from the previous
    task and the Empirical Feature Matrix to selected clients.
13:             **end if**
14:         **end if**
15:         **if** $t = 1$ **then**
16:             Train local models $\theta_j^{(t,r)}$ conventionally.             $\triangleright$ $1 \leq j \leq k$
17:         **else**
18:
19:             Train local models $\theta_j^{(t,r)}$ using Algorithm 2.
20:         **end if**
21:         Aggregate local model updates to the server.
22:     **end for**
23:     Train the data generator and the feature generator.
24:     Calculate Empirical Feature Matrix $E^t$ using synthetic features.
25: **end for**

---

**Algorithm 2** Client-side: Continual Learning

---

1: **Input:**
2: Task $T_t$, $t \geq 2$ with the dataset $\mathcal{D}_t$ in round $r$ has $B$ batches.
3: The global model $\theta_G^{(t,r)}$, a data generator $G_D^{t-1}$, a feature generator $G_F^{t-1}$.
4: The freezed global weight $\theta_G^{(t-1)}$ and the Empirical Feature Matrix $E^{t-1}$.
5: **Procedure:**
6: Calculate the Current Cross-Entropy Loss $\mathcal{L}_{\text{CE}}$ using $\mathcal{D}_t$ and Equation 9.
7: Generate synthetic data $\mathcal{D}_S$ and synthetic features $\mathcal{F}_S$ having $B$ batches each.
8: Calculate the Fine-tunig Loss $\mathcal{L}_{\text{FT}}$ using $\mathcal{D}_t$, $\mathcal{D}_S$ and Equation 11.
9: Calculate the Synthetic Logits Distillation Loss $\mathcal{L}_{\text{logits}}$ using $\mathcal{F}_S$ and Equation 10.
10: Calculate the EFM Loss $\mathcal{L}_{\text{EFM}}$ using $\mathcal{F}_S$ and Equation 12.
11: Optimize Equation 13.

---

## B    EXPERIMENTAL SETUP

In this section, we detail the settings used in our experiments, including datasets, FCIL algorithms, and experimental setups.

**Datasets** We perform our experiments on theree widely-used benchmark datasets, including the FCIL version of CIFAR-10 (Krizhevsky et al., 2009), CIFAR-100 (Krizhevsky et al., 2009) and tiny-ImageNet (Yao & Miller, 2015):

• **Sequential F-CIFAR-10.** The CIFAR-10 dataset (Krizhevsky et al., 2009) consists of 60,000 $32 \times 32$ color images in 10 classes, with 6,000 images per class. There are 50,000 training images and 10,000 test images. We split the training set into five disjoint subsets corresponding to 5 tasks.

• **Sequential F-CIFAR-100.** Sequential F-CIFAR-100 is constructed by dividing the original CIFAR-100 dataset (Krizhevsky et al., 2009), which contains 50,000 images belonging to 100 classes, into ten disjoint subsets corresponding to 10 tasks. In this way, each task has 5,000 images from 10 distinct categories, and each class has 500 images.

• **Sequential F-tiny-ImageNet.** Tiny-ImageNet (Yao & Miller, 2015) is a subset of ImageNet, containing 100,000 images of 200 real objects. We follow settings in (Babakniya et al., 2024) to form the Sequential F-tiny-ImageNet. In particular, we split the original dataset into ten non-overlapping subsets. We consider each subset as a task whose images are labeled by 20 different classes, and each class has 500 samples.

We also investigate the effectiveness of FedGTG in the context of domain shift. We introduce a protocol dataset named HealthMNIST:

• **HealthMNIST.** This dataset includes two distinct classification tasks: **Task 1** is the Colon Pathology Classifcation having data from PathMNIST (Yang et al., 2023). **Task 2** is the Blood Cell Classifcation from BloodMNIST (Yang et al., 2023). PathMNIST contains 107,180 samples of 9 classes, and BloodMNIST has 17,092 samples from 8 blood types. For both tasks, we select the first five classes from each dataset, with 500 samples per class for each task for training. Finally, the test set includes all test samples from these two datasets.

**FCIL Baselines** In addition to our FedGTG, we also include three regularization-based FCIL methods, **FLwF-2T** (Usmanova et al., 2021) and the FCIL version of **FedWeIT** (Yoon et al., 2021) and **FedEWC** (Zhang et al., 2023), and two generative-based methods, **TARGET** (Zhang et al., 2023) and **MFCL** (Babakniya et al., 2024). **FLwF-2T** utilize knowledge distillation both on the server side and client side to ease the catastrophic forgetting issue. **FedWeIT** maximizes the knowledge transfer between clients by storing previous tasks-adaptive parameters of clients. **FedEWC** is the FCIL version of EWC (Kirkpatrick et al., 2017), which uses Fisher Information Matrix (Fisher, 1922) as a regularizer to prevent forgetting. **TARGET** utilizes a global model to transfer knowledge from past tasks to the current task while also training a generator to generate synthetic data, mimicking the overall data distribution across clients. **MFCL** employs a generative model to create samples from previous distributions, which are then combined with training data to prevent catastrophic forgetting. Both of these data generation-based algorithms ensure privacy by training the generative model on the server after each task without client data retrieval.

**Implementation Details** Table 5 shows settings and the hyper-parameter tuning for each dataset.

## C GENERATIVE MODEL SETUP

**Data Generative Model Architecture** Figure 6 presents the architecture of the data generative models used for the Sequential F-CIFAR-10, Sequential F-CIFAR-100, Sequential F-tiny-ImageNet, and HealthMNIST dataset. In all experiments, the global model is based on the ResNet-18 backbone.

**Feature Generative Model Architecture** The architecure of the feature generative models is illustrated in Figure 7, which employed for the Sequential F-CIFAR-10, Sequential F-CIFAR-100, Sequential F-tiny-ImageNet, and HealthMNIST dataset. As the outputs are feature vectors, only fully connected layers are needed.

**Information generation** To create synthetic data, clients sample i.i.d. noise, which is used to determine the classes through the application of the `argmax` function to the first $q$ elements, where $q$ represents the total number of classes observed. Since the noise is sampled i.i.d., each class has an equal probability of $\frac{1}{q}$ for sample generation.

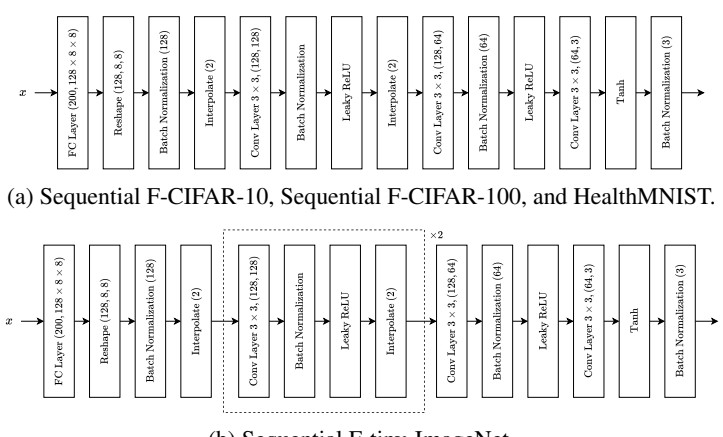

(a) Sequential F-CIFAR-10, Sequential F-CIFAR-100, and HealthMNIST.

(b) Sequential F-tiny-ImageNet.

Figure 6: Architecture of the data generative model across three datasets.

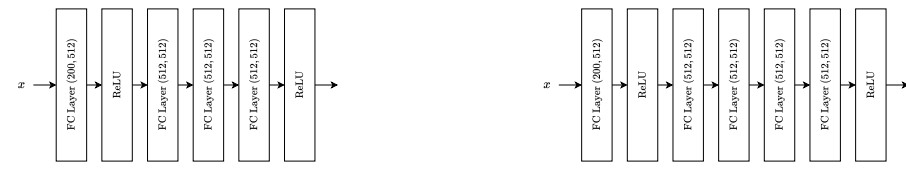

(a) Sequential F-CIFAR-10, Sequential F-CIFAR-100, and HealthMNIST.    (b) Sequential F-tiny-ImageNet.

Figure 7: Architecture of the feature generative model across three datasets.

# D  ADDITIONAL RESULTS

## D.1  TRAINING TIME COMPARISON

We use the amount of time in seconds that the clients need to complete one FL round to compare the training time between FCIL algorithms. The time is measured in rounds on our local GPU NVIDIA-A5000 and averaged between different clients. Table 4 summarizes the training time per round for all methods across various benchmarks. We can see that the increase in training time for FedGTG is comparable to MFCL and TARGET, with a moderate overhead. However, this cost is justified by the significant performance gains achieved, as demonstrated in Table 1 and Figures 3, 4, and 5 of the main paper. These results validate the efficiency and effectiveness of our approach despite the additional parameters required. We believe this balance of cost and performance underscores the practical value of FedGTG.

## D.2  ROBUSTNESS ON VARIOUS ARCHITECTURES

We conducted additional experiments using ResNet34 and ResNet50 backbones to further validate the generality and robustness of our approach. Table 6 shows the results for all methods across various datasets under the Non-IID scenario. Specifically, FedGTG consistently outperforms other FCIL algorithms across different architectures, reinforcing its adaptability and effectiveness.

## D.3  ROBUSTNESS ON CHALLENGING DATASET

Following the same setting from the work of Babakniya et al. (2024), we conducted experiments on the protocol version of ImageNet (Deng et al., 2009), named **SuperImageNet** which was created by superclassing the ImageNet dataset, thus greatly increasing the number of available samples for each class. Specifically, we conducted experiments on the SuperImageNet-L version, which consists of 7500 samples per class and 50 classes overall. The dataset was divided into 10 tasks, each of which contained 5 classes. The training process involves 300 clients, with 30 clients participating in each

| Setting | Dataset | CIFAR-10 | CIFAR-100 | tiny-ImageNet | HealthMNIST |
|---|---|---|---|---|---|
| | Image size | $32 \times 32$ | $32 \times 32$ | $64 \times 64$ | $28 \times 28$ |
| | Task number | 5 | 10 | 10 | 2 |
| | Classes per task | 2 | 10 | 20 | 5 |
| | Samples per task | 5000 | 500 | 500 | 500 |
| | Learning rate | 0.1 | 0.1 | 0.1 | 0.1 |
| Experimental Setup | Weight decay | 0.1 | 0.1 | 0.1 | 0.1 |
| | Batch size | 32 | 32 | 32 | 32 |
| | Synthetic batch size | 32 | 32 | 128 | 32 |
| | Communication round | 100 | 100 | 100 | 100 |
| | Local epoch | 10 | 10 | 10 | 10 |
| | $\lambda_{\text{IE}}$ | 1.0 | 1.0 | 1.0 | 1.0 |
| | $\lambda_{\text{batch}}$ | 1.0 | 1.0 | 1.0 | 1.0 |
| | $\lambda_{\text{smooth}}$ | 1.0 | 1.0 | 1.0 | 1.0 |
| | $\lambda_{\text{FIE}}$ | 1.0 | 1.0 | 1.0 | 1.0 |
| | $\lambda_{\text{current}}$ | 1.5 | 1.5 | 2.0 | 1.5 |
| Hyper-parameter tuning | $\lambda_{\text{FT}}$ | 1.0 | 1.0 | 1.0 | 1.0 |
| | $\lambda_{\text{logits}}$ | 0.1 | 0.1 | 0.1 | 0.1 |
| | $\lambda_{\text{EFM}}$ | 0.005 | 0.005 | 0.005 | 0.005 |
| | $\lambda_{\text{E}}$ | 10.0 | 10.0 | 10.0 | 10.0 |
| | $\eta$ | 0.1 | 0.1 | 0.1 | 0.1 |

Table 5: Detail settings across three datasets.

Table 6: Performance of the different algorithms in terms of AIA and AF for four datasets with different architectures. [↑] higher is better, [↓] lower is better.

| Method | CIFAR-10 | | CIFAR-100 | | tiny-ImageNet | | HealthMNIST | |
|---|---|---|---|---|---|---|---|---|
| | AIA (↑) | AF (↓) | AIA (↑) | AF (↓) | AIA (↑) | AF (↓) | AIA (↑) | AF (↓) |
| ResNet-34 | | | | | | | | |
| FedAvg | 42.20 | 45.68 | 21.31 | 50.41 | 12.19 | 60.94 | 61.81 | 35.51 |
| FedProx | 41.70 | 45.32 | 21.07 | 51.55 | 11.85 | 61.64 | 62.03 | 35.51 |
| FLwF-2T | 51.80 | 33.04 | 31.20 | 44.15 | 16.65 | 36.64 | 65.09 | 32.64 |
| TARGET | 57.95 | 15.98 | 42.32 | 23.20 | 19.04 | 18.27 | 66.04 | 30.74 |
| MFCL | 58.42 | 15.07 | 43.39 | 24.9 | 22.09 | 17.27 | 67.15 | 29.69 |
| **FedGTG (ours)** | **63.02** | **10.78** | **45.98** | **17.17** | **23.96** | **12.47** | **70.81** | **20.67** |
| ResNet-50 | | | | | | | | |
| FedAvg | 36.79 | 41.21 | 18.57 | 45.47 | 10.63 | 54.98 | 53.88 | 32.03 |
| FedProx | 36.35 | 40.88 | 18.37 | 46.50 | 10.33 | 55.61 | 54.08 | 32.03 |
| FLwF-2T | 45.16 | 29.81 | 27.20 | 39.82 | 14.51 | 33.06 | 56.74 | 29.45 |
| TARGET | 50.52 | 14.42 | 36.89 | 20.93 | 16.60 | 16.48 | 57.56 | 27.73 |
| MFCL | 50.93 | 13.60 | 37.82 | 22.46 | 19.26 | 15.58 | 58.54 | 26.78 |
| **FedGTG (ours)** | **54.94** | **9.73** | **40.08** | **15.49** | **20.88** | **11.25** | **61.73** | **18.64** |

round. Table 7 shows the results of various FCIL algorithms on SuperImageNet. We can see that FedGTG still outperforms other FCIL methods in this dataset, showing its ability in the field.

## D.4 ROBUSTNESS TO NATURAL CORRUPTIONS

In this section, we show additional results about the robustness of testing on natural images across our method and other FCIL methods. Figure 4 shows the last 09 augmentations of the CIFAR-100 dataset averaged over three different runs. Our approach still outperforms MFCL and TARGET in terms of test accuracy.

Table 7: Performance of the different algorithms training on the SuperImageNet dataset.

| Metrics/Method | FedGTG (ours) | MFCL | TARGET | FLwF-2T | FedWeIT | FedEWC | FedProx | FedAvg |
|---|---|---|---|---|---|---|---|---|
| AIA (↑) | **36.19** | 31.52 | 30.11 | 25.13 | 24.01 | 22.66 | 21.57 | 21.41 |
| AF (↓) | **26.08** | 32.33 | 31.56 | 41.23 | 44.36 | 49.67 | 57.99 | 59.23 |

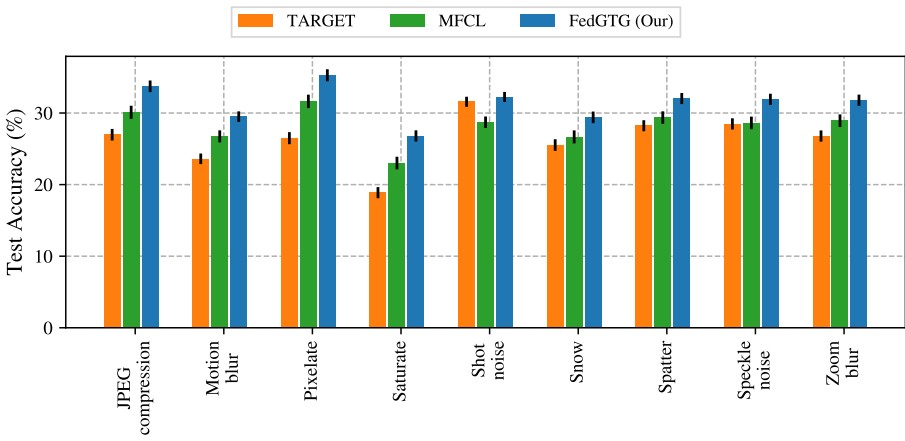

Figure 8: Robustness to natural corruptions.

## E  DATA VISUALIZATION

Figure 9 illustrates synthetic images of the CIFAR-100 dataset produced by the data generator; while these images retain specific characteristics of the original datasets, their altered shapes ensure privacy is maintained.

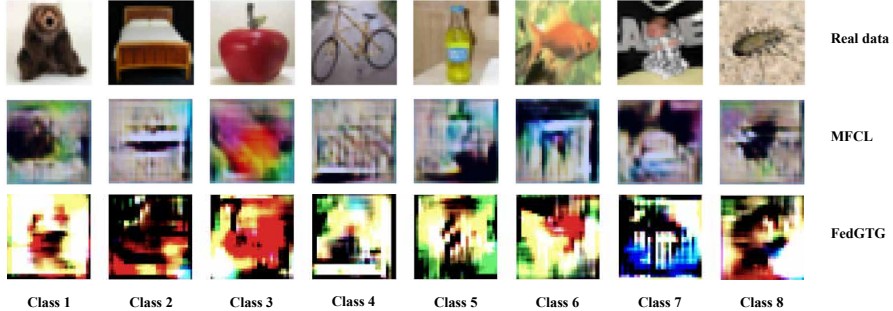

Figure 9: Synthetic images of CIFAR-100, generated by data generator from MFCL and FedGTG.

To visualize synthetic features, we employ Principal Component Analysis (Pearson, 1901) to reduce the dimensionality of the both real and synthetic features to the 2D space. As shown in Figure 10, synthetic features generated by FedGTG's feature generator align with real features, which shares the same decision boundary.

## F  COMPARISON BETWEEN FEDGTG AND MFCL

In this section, we provide a detail comparison between our framework (FedGTG) and MFCL, which both use the data generation approach to alleviate the stability-plasticity trade-off, as follows:

- In FedGTG, we additionally train the feature generator to overcome the catastrophic forgetting that MFCL still suffers from.

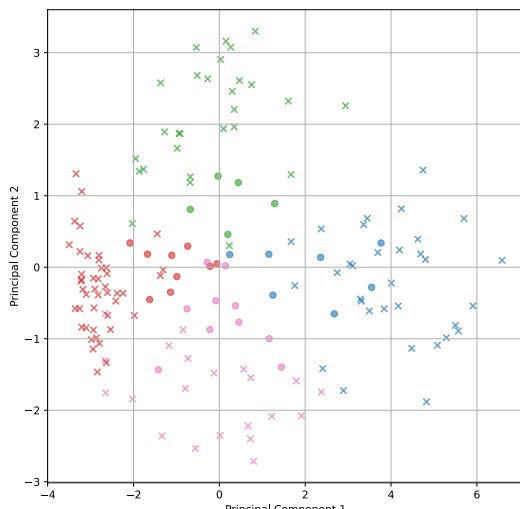

Figure 10: Synthetic features of CIFAR-100, generated by feature generator from FedGTG. $\times$ points and $\circ$ denote the real and synthetic features, respectively.

- FedGTG is effectiveness in thedomain shift scenario, where MFCL witnesses a bad stability.
- FedGTG is more robust to natural images, which is easier to interpret in the real-world scenarios.
- Although there are more computational resources needed to complete the training process on FedGTG, the increasement is comparable with MFCL, and the performance gained is better, as shown in Table 1 and Figure 3.

## G  HYPER-PARAMETER SELECTION

Hyper-parameters can have a significant impact on how well algorithms work. While it is true that each loss term in FedGTG has an associated hyper-parameter, these parameters are carefully designed to allow fine-tuning for optimal balance between knowledge retention and adaptation to new classes. We offer basic hyper-parameter settings based on extensive experimental results to help users First, note that GANs are sensitive to hyper-parameters, we set the generative model's hyper-parameters to the same values as MFCL (Babakniya et al., 2024) for a fair comparison. Second, we modify on of the local side hyper-parameters to see a difference in accuracy. Our FedGTG results from testing several hyper-parameter settings on the CIFAR-100 dataset are shown in Table 8:

Table 8: Performance of different hyper-parameters for the CIFAR-100 dataset.

| $\lambda_{FT}$ | AIA ($\uparrow$) | AF ($\downarrow$) | $\lambda_{logits}$ | AIA ($\uparrow$) | AF ($\downarrow$) | $\lambda_{EFM}$ | AIA ($\uparrow$) | AF ($\downarrow$) |
|---|---|---|---|---|---|---|---|---|
| 1.0 | 46.42 | 18.66 | 0.1 | 46.42 | 18.66 | 0.005 | 46.42 | 18.66 |
| 0.5 | 44.44 | 22.15 | 0.15 | 45.34 | 20.33 | 0.1 | 45.11 | 21.22 |
| 0.005 | 45.23 | 20.88 | 0.05 | 45.78 | 19.55 | 1 | 45.23 | 20.99 |

