# OpenReview forum: "Overcoming Catastrophic Forgetting in Federated Class-Incremental Learning via Federated Global Twin Generator"
_ICLR.cc/2025/Conference — Submitted to ICLR 2025_

### Official Review · Reviewer_aSLS · 2024-10-29

**Soundness:** 3
**Presentation:** 3
**Contribution:** 2
**Rating:** 5
**Confidence:** 4

**Summary:**

Federated Learning (FL) is a privacy-preserving machine learning approach, but it faces challenges related to resource limitations and data heterogeneity, especially as client data distributions evolve over time. Federated Class-Incremental Learning (FCIL) combines FL and Class-Incremental Learning (CIL), enabling models to continuously learn new knowledge from distributed data sources without forgetting prior knowledge. However, existing methods such as TARGET and MFCL, which use data generation to balance knowledge retention for new and old tasks, still encounter forgetting issues. To address this, this paper proposes the Federated Global Twin Generator (FedGTG) framework, which trains and shares data and feature generators on the server side, using direction-control loss to enhance stability and plasticity on the client side. Experiments demonstrate that FedGTG outperforms existing methods in terms of accuracy and forgetting rate.

**Strengths:**

1. This paper enhances current GAN methods for Federated Incremental Learning by integrating both a feature GAN and a data GAN, with the feature GAN employing regularization to achieve improved performance and stability across incremental tasks.
2. Extensive experiments across multiple datasets demonstrate the robustness and validity of the proposed model, showcasing its effectiveness in various scenarios.
3. Unlike prior GAN-based methods, which often suffer from catastrophic forgetting, this work introduces innovative mechanisms that significantly mitigate such issues, representing a valuable contribution to the field.

**Weaknesses:**

1. The experiments in the paper lack critical details, including the ratio of generated data used in each training cycle, the proportion within individual batches, and the hyperparameters for each loss function, which should ideally be explored through ablation studies. Additionally, the class-incremental setup is not clearly explained, leaving me unsatisfied with the experimental rigor.

2. The datasets used throughout the paper are relatively simple; I recommend conducting further experiments on high-resolution datasets such as ImageNet to strengthen the evaluation.

3. The contributions of the paper are limited. For instance, incorporating a feature GAN as a regularization term is not substantially different from traditional regularization methods in incremental learning.

4. It is unclear whether the GAN trained after each task is independently trained or incrementally trained based on the previous task. If trained independently, this approach would impose excessive storage and communication costs; if incrementally trained, the GAN itself could suffer from catastrophic forgetting. These aspects present limitations in the proposed method.

**Questions:**

see weakness

---

> ### Author Response · Authors · 2024-11-25
>
> We want to thank you for providing valuable comments and questions. Below is our response.
>
> ### Q1. Detailed setup
>
> Thank you for your detailed comments on the experimental setup. In our experiments, the batch size of synthetic images and features per epoch is set equal to the batch size of real images. This ensures a fair comparison with prior methods and maintains consistency across training cycles. We will make these details explicit in the revised manuscript to improve clarity.
>
> In the class-incremental setup, we conducted experiments using three widely recognized benchmarks for class-incremental learning as follows:
>
> - Sequential F-CIFAR-10: The CIFAR-10 dataset consists of 60,000 32 × 32 color images in 10 classes, with 6,000 images per class. There are 50,000 training images and 10,000 test images. We split the training set into five disjoint subsets corresponding to 5 tasks.
> - Sequential F-CIFAR-100. Sequential F-CIFAR-100 is constructed by dividing the original CIFAR-100 dataset, which contains 50,000 images belonging to 100 classes, into ten disjoint subsets corresponding to 10 tasks. In this way, each task has 5,000 images from 10 distinct categories, and each class has 500 images.
> - Sequential F-tiny-ImageNet. Tiny-ImageNet is a subset of ImageNet, containing 100,000 images of 200 real objects. We follow settings in the work of Babakniya et al. [1] to form the Sequential F-tiny-ImageNet. In particular, we split the original dataset into ten nonoverlapping subsets. We consider each subset as a task whose images are labeled by 20 different classes, and each class has 500 samples.
>
> These experimental setups were detailed in the Appendix, but we agree they are critical for understanding and will include them in the main text upon acceptance.
>
> ### Q2. Experiments on challenging dataset
>
> Thank you for your valuable suggestion to evaluate our approach on high-resolution datasets such as ImageNet. We agree that testing on ImageNet would provide stronger validation of the scalability and effectiveness of our method in handling complex and diverse real-world scenarios. Therefore, following the same setting from MFCL [1], we conducted experiments on the protocol version of ImageNet, named SuperImageNet where a dataset created by superclassing the ImageNet dataset, thus greatly increasing the number of available samples for each class. Below are the results of various FCIL algorithms on this dataset:
>
> | Method  | AIA   | AF    |
> |---------|-------|-------|
> | FedAvg  | 21.41 | 59.23 |
> | FedProx | 21.57 | 57.99 |
> | FLwF-2T | 25.13 | 41.23 |
> | TARGET  | 30.11 | 31.56 |
> | MFCL    | 31.52 | 32.23 |
> | FedGTG (Our) | 36.19 | 26.08 |
>
> We can see that FedGTG still outperforms other FCIL methods in this dataset, showing its ability in the field.
>
> ### Q3. Contribution concerns
>
> Thank you for your feedback regarding the contributions. While incorporating a feature GAN might seem similar to traditional regularization methods, our novelty lies in the dual-generator architecture, which includes both data and feature generators. This dual approach addresses two critical challenges in federated class-incremental learning:
>
> - Catastrophic Forgetting: The feature generator effectively preserves knowledge across tasks by balancing stability and plasticity, reducing bias toward recent classes.
> - Bias Correction: The data generator complements the feature generator by ensuring robustness and better generalization across tasks, mitigating class imbalance.
>
> These contributions are supported by extensive analyses and real-world experimental results, which demonstrate practical advantages over existing techniques. Additionally, our method introduces new insights into combining generative models and regularization for improved knowledge retention, which we believe distinguishes it from prior work.
>
> ### Q4. Generative model concerns
>
> Thank you for pointing out this important question. Our approach employs a data-free training mechanism for the generative model [2, 3], guided by the global model. Specifically, the generative model produces synthetic data based on the knowledge encoded in the global model after each task. This process does not require independent training for each task, thereby avoiding excessive storage or communication costs.
>
> We acknowledge the potential issue of catastrophic forgetting within the generative model itself if trained incrementally. To address this, we focus on ensuring the global model effectively preserves existing knowledge and adapts to new tasks. This foundation improves the quality of data generated by the GAN. Preventing catastrophic forgetting in the generative model remains an open challenge and will be an area of focus for future studies.

---

### Official Review · Reviewer_NVr3 · 2024-10-29

**Soundness:** 3
**Presentation:** 3
**Contribution:** 3
**Rating:** 5
**Confidence:** 4

**Summary:**

- This paper focuses on federated continual learning, federated class-incremental learning to be more specific. It proposes a new method of using two generator to generate synthetic data and feature. The synthetic data and feature are sent to the client for local training. In local training, this work proposes multiple losses to improve the training process.

**Strengths:**

- The propose method of using an additional generator is simple and intuitive, while it is not straightforward to make it work. It is technically sound that the authors propose additional loss functions to stablize the training.
- The paper is generally well-written and easy to follow.
- Extensive experiments are conducted through 3 datasets with detailed ablation studies. Sufficient exisiting works are compared. The proposed method achieves much strong performance than the compared counterparts.

**Weaknesses:**

- Figure 1 is not easy to  understand. It would be useful if the authors could provide more explanations or making it more intuitive.
- The abstract claims that “these approaches’ testing accuracy in previous classes is still much lower than recent classes”, but it seems the paper does not provide experiments showing that the proposed method addresses this issue.
- It seems that only a single backbone is used in the evaluatiion.
- Typo in line 83, should be “methods”.

**Questions:**

- Does this work focus on cross-silo or cross-device FL? What is the number of clients evaluated?
- What is the meaning of “stability-plasticity” ?

---

> ### Author Response · Authors · 2024-11-25
>
> We appreciate the comments and suggestions from the reviewer. The following is our response to the raised concerns.
>
> ### Q1. Explaining Figure 1
>
> Thank you for your observation. We agree that Figure 1 could be made more intuitive. Our approach is inspired by foundational works such as TARGET [17] and MFCL [19], which utilize generative models in federated class-incremental learning (FCIL). To clarify, Figure 1 illustrates that using only a data generator leads to bias towards recent classes, as shown in Figure 2 of the paper. To address this issue, we introduced a feature generator (Section 3.2.2), which enhances the balance between stability (retaining past knowledge) and plasticity (learning new tasks).
>
> We have updated Figure 1 to include clearer labels, step-by-step explanations, visual cues to highlight the interplay between the two generators and the additional loss terms that mitigate these challenges.
>
> ### Q2. Previous classes accuracy concern
>
> Thank you for pointing this out. The imbalance between learning new tasks and retaining old knowledge is a well-recognized issue in continual learning, referred to as the stability-plasticity dilemma. To address this, we introduced a feature generator that improves the model’s ability to retain knowledge from earlier tasks, thereby increasing the testing accuracy in previous classes.
>
> Our experimental results in Table 1 demonstrate that FedGTG achieves the lowest Average Forgetting (AF) among the evaluated FCIL methods, showcasing its ability to effectively preserve knowledge from past tasks. This result directly supports our claim in the abstract. We have revised the manuscript to highlight this evidence more clearly to ensure alignment with the claims made in the abstract.
>
> ### Q3. Backbone evaluation
>
> Thank you for raising this concern. In response, we conducted additional experiments using ResNet34 and ResNet50 backbones to further validate the generality and robustness of our approach. Below are the updated results for all methods across various datasets under the Non-IID scenario:
>
> **Table R1: Performance of various FCIL algorithms using ResNet34.**
>
> |Dataset|CIFAR-10|CIFAR-10 |CIFAR-100|CIFAR-100 |tiny-ImageNet| tiny-ImageNet|HealthMNIST|HealthMNIST |
> |:---:|:---:|:---:|:---:|:---:|:---:|:---:|:---:|:---:|
> |Method |AIA|AF|AIA|AF|AIA|AF|AIA|AF|
> |FedAvg|42.20|45.68|21.31|50.41|12.19|60.94|61.81|35.51|
> |FedProx|41.70|45.32|21.07|51.55|11.85|61.64|62.03|35.51|
> |FLwF-2T|51.80|33.04|31.20|44.15|16.65|36.64|65.09|32.64|
> |TARGET|57.95|15.98|42.32|23.20|19.04|18.27|66.04|30.74|
> |MFCL|58.42|15.07|43.39|24.90|22.09|17.27|67.15|29.69|
> |FedGTG|63.02|10.78|45.98|17.17|23.96|12.47|70.81|20.67|
>
> **Table R1: Performance of various FCIL algorithms using ResNet50.**
>
> |Dataset|CIFAR-10|CIFAR-10 |CIFAR-100|CIFAR-100 |tiny-ImageNet| tiny-ImageNet|HealthMNIST|HealthMNIST |
> |:---:|:---:|:---:|:---:|:---:|:---:|:---:|:---:|:---:|
> |Method |AIA|AF|AIA|AF|AIA|AF|AIA|AF|
> |FedAvg|36.79|41.21|18.57|45.47|10.63|54.98|53.88|32.03|
> |FedProx|36.35|40.88|18.37|46.50|10.33|55.61|54.08|32.03|
> |FLwF-2T|45.16|29.81|27.20|39.82|14.51|33.06|56.74|29.45|
> |TARGET|50.52|14.42|36.89|20.93|16.60|16.48|57.56|27.73|
> |MFCL|50.93|13.60|37.82|22.46|19.26|15.58|58.54|26.78|
> |FedGTG|54.94|9.73|40.08|15.49|20.88|11.25|61.73|18.64|
>
> As shown in Table R1 and Table R2, FedGTG consistently outperforms other FCIL algorithms across datasets, reinforcing its adaptability and effectiveness.
>
> ### Q4. Typo in line 83, should be “methods”.
>
> Thank you for pointing out the typo. We have corrected it in the revised manuscript.
>
> ### Q5. Does this work focus on cross-silo or cross-device FL? What is the number of clients evaluated?
>
> Thank you for your question. Our approach is flexible and can be applied to both cross-silo and cross-device FL scenarios. In our experiments, we set the default number of clients to 50 to simulate a realistic FL environment.
>
> To evaluate the robustness of FedGTG across varying client sizes, we performed additional analyses, as presented in Figures 5e and 5f of the paper. These figures demonstrate that FedGTG maintains its superior performance across a range of client sizes, showcasing its adaptability to different FL setups. We have clarified this in the revised manuscript to better highlight the applicability of our work.
>
> ### Q6. What is the meaning of “stability-plasticity”?
>
> In continual learning, the stability-plasticity dilemma refers to the challenge of achieving a balance where a model remains stable enough to retain previously learned knowledge (stability) while also being adaptable to learn new information (plasticity).
>
> An ideal continual learning system must minimize catastrophic forgetting (loss of old knowledge) while maintaining sufficient flexibility to acquire new information. In our work, this balance is achieved through the dual-generator structure, where the feature generator improves stability and the data generator ensures sufficient plasticity for learning new tasks.

---

### Official Review · Reviewer_1TrM · 2024-11-01

**Soundness:** 3
**Presentation:** 2
**Contribution:** 3
**Rating:** 5
**Confidence:** 4

**Summary:**

This paper introduces Federated Global Twin Generator (FedGTG), and proposes a framework leveraging generative models on the global side, sending synthetic data to clients for retaining knowledge and learning new tasks effectively. FedGTG shows enhanced accuracy, reduced forgetting, and robustness against domain shifts on CIFAR-10, CIFAR-100, and tiny-ImageNet.

**Strengths:**

1. The paper is overall well-written in terms of writing.
2. The introduction part clearly especify the problem about generators.
3. The experiments are showing nontrivial improvements over compared techniques.

**Weaknesses:**

1. Not clear what the biggest contributions are. Is it the two generator structure? If so, the novelty does not seem very high. The paper also introduces a direction-controlling objective, which should also stand out, but not very clearly in the paper (e.g., Figure 1).
2. Need to include training time or cost in the comparison. It seems that training two generators are quite costing, and it already needs more parameter during the training.
3. Is the method an exemplar-free technique? If so, the literature reviewer is not very complete. Quite a few new works on exemplar-free CIL are missing in the continual learning section, such as the analytic continual learning branch [1-3], and prototypes-based CIL.

[1] "ACIL: Analytic class-incremental learning with absolute memorization and privacy protection." Advances in Neural Information Processing Systems 35 (2022): 11602-11614.

[2] "GKEAL: Gaussian kernel embedded analytic learning for few-shot class incremental task." Proceedings of the IEEE/CVF Conference on Computer Vision and Pattern Recognition. 2023.

[3] "DS-AL: A Dual-Stream Analytic Learning for Exemplar-Free Class-Incremental Learning." Proceedings of the AAAI Conference on Artificial Intelligence. Vol. 38. No. 15. 2024.

**Questions:**

see weaknesses.

---

> ### Author Response · Authors · 2024-11-25
>
> Thank you for taking the time and effort to review our paper. We appreciate your valuable feedbacks and comments. We would like to address your concerns as follows:
>
> ### Q1. Contribution concerns
>
> Thank you for raising this concern. The key contribution of our work lies in the novel integration of dual generators (a data generator and a feature generator) collaboratively address catastrophic forgetting and mitigate the bias problem towards recent classes in federated class-incremental learning (FCIL). While the two-generator structure is an important aspect, the direction-controlling objective is equally significant, as it regulates the alignment of learned features across tasks, thereby enhancing model generalization.
>
> To improve clarity, we have revised the manuscript to emphasize the role of the direction-controlling objective more explicitly. For instance, Figure 1 has been updated to better illustrate how the objective interacts with the dual generators and its impact on the learning process. This combination of mechanisms provides practical advantages, which are further validated through comprehensive experiments that demonstrate superior performance compared to existing techniques.
>
>
> ### Q2. Training time comparison
>
>
> We appreciate this valuable suggestion. To address this, we have included a detailed analysis of training time and computational costs in the revised manuscript. The table below summarizes the training time per round for all methods across various benchmarks:
>
> |Training Time/Method|FedGTG (Our)|MFCL|TARGET|FLwF-2T|FedProx|FedAvg|
> |:---:|:---:|:---:|:---:|:---:|:---:|:---:|
> |Task = 1|1.2|1.2|1.2|1.2|1.8|1.2|
> |Task > 1|4.1|3.7|3.5|3.4|1.8|1.2|
>
> As shown above, the increase in training time for FedGTG is comparable to MFCL and TARGET, with a moderate overhead. However, this cost is justified by the significant performance gains achieved, as demonstrated in Table 1 and Figures 4, 5, and 6 of the main paper. These results validate the efficiency and effectiveness of our approach despite the additional parameters required. We believe this balance of cost and performance underscores the practical value of FedGTG.
>
> ### Q3. Missing related works
>
> Thank you for pointing out this oversight. Yes, our method is exemplar-free, as it avoids storing past data while effectively tackling class-incremental learning challenges. We have revised the literature review section to include key works in exemplar-free continual learning, including the analytic continual learning branch [1-3] and prototypes-based CIL. These additions ensure a more comprehensive contextualization of our approach within the broader field.

---

### Official Review · Reviewer_6TCk · 2024-11-02

**Soundness:** 2
**Presentation:** 4
**Contribution:** 2
**Rating:** 5
**Confidence:** 4

**Summary:**

This paper introduce FedGTG (Federated Global Twin Generator), a novel FCIL framework that leverages generative models on the server side without accessing client data. FedGTG trains a data generator and a feature generator to produce synthetic representations of all seen classes, which are then shared with clients to help balance knowledge retention (stability) and learning new tasks (plasticity). Through extensive experiments on CIFAR-10, CIFAR-100, and tiny-ImageNet, FedGTG shows significant improvements in both accuracy and resistance to forgetting

**Strengths:**

1. The paper writing is clear and easy to follow.
2. The proposed FedGTG absorbs the advantage of many other algorithms, leading to a strong framework for FCIL.
3. The experimental results are good. The proposed FedGTG outperforms all baselines on each benchmark and each incremental task. And the ablation study shows that each loss term is crucial and  has a large impact on the final results.

**Weaknesses:**

1. The proposed lacks enough novelty. All loss functions used in FedGTG are proposed by other papers.
2. The framework contains too many hyperparameters, as each loss term needs one. It seems the framework is hard to be finetuned to the optimal result, as there are too many combinations of choice of hyperparameters. And the paper lacks the experiments and discussion of these hyperparameters.
3. More baselines should be compared. Except for generative-based methods, authors should compare FedGTG with other SOTA FCIL algorithms.

**Questions:**

Why do you design the model architecture of data and feature generation as you shown in the paper? Are there any key ideas of why using these architectures?

---

> ### Author Response · Authors · 2024-11-25
>
> Thank you for your positive feedback and constructive comments about our contributions. We have address your questions and comments below. Please let us know if you have any remaining questions.
>
> ### Q1. Novelty concerns
> Thank you for your feedback regarding novelty. While we acknowledge that the individual loss functions used in FedGTG are not novel, our contribution lies in the integration of dual generators (a data generator and a feature generator) into a unified framework. This approach specifically addresses the challenge of catastrophic forgetting in federated class-incremental learning (FCIL) and mitigates the bias towards recent classes. Additionally, our extensive analysis validates the effectiveness of this dual-generator approach across various real-world scenarios, offering practical insights and demonstrating significant improvements over existing techniques. We believe this innovative integration and its application to FL highlight the novelty of our work.
>
> ### Q2. Hyperparameter selection
>
> We appreciate your concern about the number of parameters. While it is true that each loss term in FedGTG has an associated hyperparameter, these parameters are carefully designed to allow fine-tuning for optimal balance between knowledge retention and adaptation to new classes. We offer basic hyperparameter settings based on extensive experimental results to help users First, note that GANs are sensitive to hyperparameters, we set the generative model's hyperparameters to the same values as MFCL [1] for a fiar comparison. Second, we modify one of the local side hyperparameters to see a difference in accuracy. Our FedGTG results from testing several hyperparameter settings on the CIFAR-100 dataset are shown below:
>
> **Table H1: Performance of FedGTG through various $\lambda_{\text{FT}}$.**
>
> | $\lambda_{\text{FT}}$   | AIA   | AF    |
> |-------|-------|-------|
> | 1.0   | 46.42 | 18.66 |
> | 0.5   | 44.44 | 22.15 |
> | 0.005 | 45.23 | 20.88 |
>
> **Table H2: Performance of FedGTG through various $\lambda_{\text{logits}}$.**
>
> | $\lambda_{\text{logits}}$   | AIA   | AF    |
> |-------|-------|-------|
> | 0.1   | 46.42 | 18.66 |
> | 0.15  | 45.34 | 20.33 |
> | 0.05  | 45.78 | 19.55 |
>
> **Table H3: Performance of FedGTG through various $\lambda_{\text{EFM }}$.**
>
> | $\lambda_{\text{EFM}}$   | AIA   | AF    |
> |-------|-------|-------|
> | 0.005 | 46.42 | 18.66 |
> | 0.1   | 45.11 | 21.22 |
> | 1     | 45.23 | 20.99 |
>
> Table H1-3 shows that FedGTG is insensitive to hyperparameter settings, making it simple for users to select the ideal set of hyperparameters for their tasks. We have added these results to the revised manuscript to provide a more comprehensive evaluation.
>
> ### Q3. Baselines comparison
>
> In our manuscript, we compare FedGTG with a regularizer-based approach named FLwF-2T. For more baselines comparison, we conducted experiments on the federated class-incremental version of two regularized-based methods, FedWeIT [4] and FedEWC [5]. Below are the results for all methods across various datasets under the Non-IID scenario ($\alpha=0.5$) using ResNet18:
>
> **Table M1: Performance of various FCIL algorithms using ResNet18.**
>
> |Dataset|CIFAR-10|CIFAR-10 |CIFAR-100|CIFAR-100 |tiny-ImageNet| tiny-ImageNet|HealthMNIST|HealthMNIST |
> |:---:|:---:|:---:|:---:|:---:|:---:|:---:|:---:|:---:|
> |**Method** |**AIA**|**AF**|**AIA**|**AF**|**AIA**|**AF**|**AIA**|**AF**|
> | FedAvg  | 40.92 | 55.59 | 20.66 | 61.34 | 11.82 | 74.16 | 59.93 | 43.21 |
> | FedProx | 40.43 | 55.15 | 20.43 | 62.73 | 11.49 | 75.01 | 60.15 | 43.21 |
> | FedEWC  | 43.22 | 50.7  | 25.43 | 59.17 | 12.9  | 60.93 | 60.88 | 42.45 |
> | FedWeIT | 48.11 | 47.34 | 28.89 | 56.11 | 14.55 | 49.76 | 61.05 | 42.09 |
> | FLwF-2T | 50.23 | 40.21 | 30.25 | 53.72 | 16.14 | 44.59 | 63.11 | 39.72 |
> | TARGET  | 56.19 | 19.45 | 41.03 | 28.23 | 18.46 | 22.23 | 64.03 | 37.41 |
> | MFCL    | 56.65 | 18.34 | 42.07 | 30.3  | 21.42 | 21.02 | 65.11 | 36.13 |
> | FedGTG  | 61.11 | 13.12 | 44.58 | 20.89 | 23.23 | 15.18 | 68.66 | 25.15 |
>
> As shown in Table M1, the performance of FedGTG still outperforms other FCIL methods, fostering its effectiveness in the FCIL setting.
>
> ### Q4. Architecture design
>
> Thank you for your question regarding the architecture design. The architecture for the data generation model was adopted from [1] to maintain consistency with existing benchmarks, enabling a fair comparison with prior techniques. For the feature generation model, we utilized fully connected layers, as they are well-suited for processing flattened feature vectors, ensuring computational efficiency and scalability. The design choice aligns with our goal of balancing effective knowledge generation and resource constraints inherent to FL environments. We believe these architectural decisions provide a strong foundation for evaluating FedGTG while addressing the unique requirements of FCIL tasks.

---

> > ### Comment · Reviewer_6TCk · 2024-11-25
> >
> > Thanks to the authors for their rebuttal, which addresses some of my concerns. I have raised my score. However, I still believe that integrating existing works cannot be considered as a significant contribution to the FL community.

---

> > > ### Author Response · Authors · 2024-11-30
> > >
> > > Thank you very much for your valuable comments. We greatly appreciate your time in reviewing our work. We'd like to further clarify our answers below.
> > >
> > > Dual generators [6, 7] are implemented in standard FL settings, but this concept was not previously applied in FCIL. For the first time, we have integrated dual generators (a data generator and a feature generator) into a unified framework, which is specifically designed to tackle the issue of catastrophic forgetting in FCIL and to reduce bias toward more recent classes.
> > >
> > > Furthermore, according to the survey "Federated Continual Learning for Edge-AI: A Comprehensive Survey" [8], existing generative-replay-based FCIL algorithms have primarily focused on addressing catastrophic forgetting, instead of taking the practicality of the method into account. To the best of our knowledge, our paper firstly proposed the analysis on the implementation of FCIL algorithms in real-world scenarios. Specifically, we firstly:
> > >
> > > 1. Conducting initial experiments to assess the adaptability of FCIL algorithms to domain shifts.
> > > 2. Evaluating the robustness of FCIL algorithms in natural settings, using the CIFAR-100-C dataset as a benchmark.
> > > 3. Testing the calibration and generalization capabilities of FCIL methods.
> > >
> > > As shown in Table 1 and Figure 5-6, FedGTG outperforms others in terms of Average Incremental Accuracy (AIA) and Average Forgetting (AF), proving its practicality in the real-world setting.
> > >
> > > In summary, we believe that our work is emphasized by the novelty of this innovative combination and its implementation in real-world FL.
> > >
> > > ### References:
> > >
> > > [6] Luo, Kangyang, et al. "DFDG: Data-Free Dual-Generator Adversarial Distillation for One-Shot Federated Learning." arXiv preprint arXiv:2409.07734 (2024).
> > >
> > > [7] Ma, Zhuoran, et al. "Flgan: Gan-based unbiased federated learning under non-IID settings." IEEE Transactions on Knowledge and Data Engineering (2023).
> > >
> > > [8] Wang, Zi, et al. "Federated Continual Learning for Edge-AI: A Comprehensive Survey." arXiv preprint arXiv:2411.13740 (2024).

---

### Author Response · Authors · 2024-11-25

We want to thank you the reviewers for providing constructive comments and valuable questions, helping us to improve our paper's quality. In our rebuttal, we have included new experiments in response to all of the reviewers' questions and recommendations. We hope all reviewers to provide any further feedback they may have. We promise to respond to any further inquiries as soon as possible. Lastly, we respectfully ask for the re-evaluation the score assigned to our paper. We really appreciate your help.

### Main changes

1. We added training time comparison between FedGTG and other methods. The training time of FedGTG is comparable with MFCL but achieved better performance, as shown in Table 1, Figure 4-6.
2. We conducted the experiment on different architectures, including ResNet34 and ResNet50 on CIFAR-10, CIFAR-100, tiny-ImageNet and HealthMNIST, as well as on additional regularized-based algorithms, which are FedWeIT and FedEWC. Experimental results have shown that our method can retain old knowledge and learn new tasks efficiently, while the two prior works fail to do so.
3. We conducted additional dataset which is SuperImageNet to test the effectiveness of FedGTG under a more challenging scenario.
4. We addressed hypeparameter selection to help users fine-tune their work easier.
5. We address minor typos in the manuscripts.

### References

[1] Babakniya, Sara, et al. "A data-free approach to mitigate catastrophic forgetting in federated class incremental learning for vision tasks." Advances in Neural Information Processing Systems 36 (2024).

[2] Yin, Hongxu, et al. "Dreaming to distill: Data-free knowledge transfer via deepinversion." Proceedings of the IEEE/CVF conference on computer vision and pattern recognition. 2020.

[3] Smith, James, et al. "Always be dreaming: A new approach for data-free class-incremental learning." Proceedings of the IEEE/CVF international conference on computer vision. 2021.

[4] Yoon, Jaehong, et al. "Federated continual learning with weighted inter-client transfer." International Conference on Machine Learning. PMLR, 2021.

[5] Zhang, Jie, et al. "Target: Federated class-continual learning via exemplar-free distillation." Proceedings of the IEEE/CVF International Conference on Computer Vision. 2023.

---

### Meta-Review · Area_Chair_Ao4z · 2024-12-20

**Metareview:**

This paper introduces FedGTG (Federated Global Twin Generator), a framework that leverages generative models on the server side without accessing client data. The main idea is quite simple, using two generators to generate synthetic data and features, which are sent to the client for local training. One common concern shared by multiple reviewers is the novelty of the paper. In particular, as pointed out by reviewers, the main loss function design for local training boils down to a combination of existing losses in the literature, and it lacks principled justification on why one should do so. The authors' responses answered some of the clarification questions from the reviewers but the key limitation still remains. The reviewers have reached a consensus of not accepting the paper.

**Additional Comments On Reviewer Discussion:**

Reviewers are in consensus.

---

### Decision · Program_Chairs · 2025-01-22

Reject